# Established and Evolving Roles of the Multifunctional Non-POU Domain-Containing Octamer-Binding Protein (NonO) and Splicing Factor Proline- and Glutamine-Rich (SFPQ)

**DOI:** 10.3390/jdb12010003

**Published:** 2024-01-05

**Authors:** Danyang Yu, Ching-Jung Huang, Haley O. Tucker

**Affiliations:** 1Department of Biology, New York University in Shanghai, Shanghai 200122, China; dy276@nyu.edu; 2Molecular Biosciences, Institute for Cellular and Molecular Biology, University of Texas at Austin, 1 University Station A5000, Austin, TX 78712, USA

**Keywords:** NonO/p54nrb, SFPQ/PSF, DNA damage response, ATR/ATM, cell cycle, checkpoint control, birth defects, cancer

## Abstract

It has been more than three decades since the discovery of multifunctional factors, the Non-POU-Domain-Containing Octamer-Binding Protein, NonO, and the Splicing Factor Proline- and Glutamine-Rich, SFPQ. Some of their functions, including their participation in transcriptional and posttranscriptional regulation as well as their contribution to paraspeckle subnuclear body organization, have been well documented. In this review, we focus on several other established roles of NonO and SFPQ, including their participation in the cell cycle, nonhomologous end-joining (NHEJ), homologous recombination (HR), telomere stability, childhood birth defects and cancer. In each of these contexts, the absence or malfunction of either or both NonO and SFPQ leads to either genome instability, tumor development or mental impairment.

## 1. Introduction 

NonO was first reported more than three decades ago [1,2,3] as a single-strand octamer DNA-binding protein and, subsequently, as a DNA- and RNA-binding protein [2]. SFPQ was first purified with another RNA-binding protein, Polypyrimidine Tract Binding (PTB) [4], and defined as a splicing factor [5]. SFPQ participates in numerous mechanisms, including transcription (reviewed in [6,7]), posttranscription ([8,9]; reviewed in [10]) and RNA processing [11,12]. Early studies of NonO and SFPQ also focused on their direct regulation of transcription [13], posttranscription [14] and 3′ end pre-mRNA processing [15]. Subsequently, NonO and SFPQ were found to be key components of nuclear paraspeckles, which control gene expression through the nuclear retention of RNA ([16,17]; reviewed recently in [18]). 

NonO and SFPQ (Figure 1) are members of the Drosophila Behavior/Human Splicing (DBHS) family of RNA/DNA-binding cofactors, which are liable for a range of cellular processes (reviewed in [19,20,21]). The DBHS family also includes PSPC1, Hrp65 [22], NonO-1 (in *C. elegans* ([23]) and NonA [24] (Figure 2A). DBHS family members share a common internal modular organization and bind DNA and RNA via two consecutive, highly conserved RNA-recognition motifs (RRMs). Most DBHS proteins also include a NOPS (NonA/paraspeckle) domain and coiled coil regions which extend out from the core of the dimer to form an alpha-helical module required for protein polymerization (reviewed in [21]). Thus, the overall structures of NonO, SFPQ and PSPC1/PSP1 share over 70% identity at the protein level. They are particularly similar in the regions required for the formation of their homo-/hetero-dimers [25,26,27]. These protein-sequence-based observations were validated at the atomic level via several X-ray structure analyses (for example, [28,29,30,31,32,33]).

## 2. Multifunctional Roles of NonO and SFPQ

The studies above, as well as the others described below, established NonO and SFPQ as “multifunctional” proteins (reviewed in [6]). For example, NonO is involved in cell proliferation ([26]; reviewed in [34]), whereas SFPQ binds U5 small nuclear RNA and participates in splicing [35]. NonO, SFPQ and PSPC1 are key components of paraspeckles (reviewed in [21]). To mediate this function, NonO interacts with CARM1 (histone coactivator-associated arginine methyltransferase 1) [36] to localize CARM1 within paraspeckles for its regulation in pre-implantation mouse embryonic development [37]. NonO and SFPQ can act as RNA splicing regulators, and each has been implicated in neurogenesis (reviewed in [38]), differentiation and development [39,40]. The loss of NonO or SFPQ results in apoptosis and senescence ([41]; reviewed in [7]). 

NonO and SFPQ can function individually, but often as heterodimers or as larger complexes [26,27]. For example, NonO, but not SFPQ, plays a significant role in circadian rhythm [42,43]. The disruption of NonO and SFPQ heterotetramers induces premature senescence [27]. The dysregulation of either NonO or SFPQ, or both, can lead to tumorigenesis [44,45]. For example, NonO upregulation can lead to malignant breast cell proliferation [46], whereas NonO and SFPQ together promote castration-resistant prostate cancer progression [47,48]. The fusion of either NonO or SFPQ with the TFE3 transcriptional factor has been observed in papillary renal cell carcinoma [49,50]. This same SFPQ–TFE3 oncoprotein inactivates TFE3 and p53 in papillary renal cell carcinoma [51] (Figure 2B).

The loss-of-function variants of NonO were previously associated with intellectual disability in males [52], yet follow-up studies were not performed at that time. However, recently, pathogenic hemizygous forms of NonO were shown to result in a rare X-linked syndromic disorder. The inflicted males showed development delay, corpus callosum anomalies and macrocephalic birth defects [53]. As yet, there remains no role for SFPQ in these aforementioned “NonO-associated syndromic disorders”, although SFPQ disruption was shown to result in neuronal apoptosis in developing mouse brains [54].

## 3. Focus of This Review

In addition to the multifunctional roles described above, NonO and SFPQ participate in double-strand break repair in response to DNA damage [55,56]. Observations made several years ago showed that both NonO and SFPQ function in DNA unwinding and pairing [56]. In this review, we focus on these roles as well as how the interactions among NonO and/or SFPQ with other DNA damage repair factors illuminate their dual roles in double-strand break (DSB) repair. We discuss the recent revelations of NonO in birth defects and conclude with a more detailed consideration of NonO and SFPQ in cancer.

## 4. NonO and SFPQ in Cell Cycle and Cell Cycle Arrest

Long ago, a dimeric complex of NonO:SFPQ was shown to be more highly proliferative in cancer cell lines than in normal cells [26]. NonO silencing induced a reduction in the G1/S phase. Also, NonO was shown to be required for the arrest of DNA replication (S phase) with ultraviolet-C (UVC) [57]. After UVC treatment, NonO was localized with Rad9, an ortholog of 53BP1, foci [57]. These and other data indicated that NonO acts upstream of the ATR-mediated DNA damage response cascade [57]. NonO is required for responding to UV-irradiation-mediated checkpoint arrest. The tyrosine phosphorylation of SFPQ, on the other hand, has been reported to promote S-phase cell cycle arrest and its cytoplasmic localization [58]. 

NonO and SFPQ are also required for G2/M arrest [59,60]. In search of G2/M-arrest-related proteins, Roberts et al. [61] identified NonO as one of two protein targets phosphorylated by MKK 1/2 inhibitors [27,61]. Microtubule-interfering agents and other agents, such as kinesin spindle protein (KSP) inhibitors, were required to induce NonO phosphorylation [62]. Similar approaches revealed that SFPQ is hyperphosphorylated in G2/M arrest [63]. The phosphorylation of NonO or SFPQ occurs during cell cycle arrest.

In summary, both NonO and SFPQ are required to arrest the cell cycle, and phosphorylation of either or both might be involved in triggering the cell cycle for downstream regulation. 

## 5. NonO and SFPQ Participate in DNA Damage Repair

The initial indication that NonO was required for DNA damage repair came from observations by Roberts et al. [61]. Following purification during G2/M arrest, NonO was observed to be phosphorylated. Also, SFPQ had previously been shown to be hyperphosphorylated during G2/M arrest [56]. These studies indicated that NonO and SFPQ phosphorylation are associated with cell cycle arrest. Further studies indicated that following cellular stress or UV-induced damage, NonO and SFPQ participate in double-strand break (DSB) repair [56].

The two major pathways of DNA double-strand break (DSB) repair are nonhomologous end-joining (NHEJ) and homologous recombination (HR) (reviewed in [64]. NHEJ is the primary pathway for repairing DNA double-strand breaks (DSBs) (reviewed in [65] and summarized by the authors in their Figure 1). Briefly, during the S and G2 phases, homology-directed repair is typical and includes homologous recombination (HR) and single-strand annealing (SSA). At any time in the cell cycle, double-stranded breaks are subject to repair by nonhomologous DNA end-joining (NHE). Of the numerous proteins required, a key role in NHEJ is provided by Mre11-Rad50-NBS1 (MRN) (further discussed below). 

NonO and SFPQ were isolated as a previously uncharacterized ~200 kDa fraction of MRN (detailed in Figure 1 of [55]). Subsequently, the complex was found to be devoid of the crucial ATP-dependent DNA ligase activity required to covalently join the adjacent 5′ phosphate (5′P) and 3′ hydroxy (3′OH) termini of double-stranded DNA [56]. The attenuation of NonO led to cell cycle arrest [66] and delayed DNA damage repair, whereas radiation-induced chromosomal aberrations increased [66]. 

These studies concluded that the mechanism employed by the NonO:SFPQ heterodimer is to promote the binding of DNA substrates and the stimulation of DNA end-joining. This activity, in cooperation with Ku, leads to the formation of the “preligation complex” [56].

## 6. NonO and SFPQ Participate in Nonhomologous End-Joining (NHEJ)

There are several key polypeptides required for NHEJ: Two subunits of Ku, which bind to DNA ends; DNA ligase IV (L4) and XRCC4 (X4), which form a complex that catalyzes strand ligation; and DNA-PKcs, the only active protein kinase described in the NHEJ pathway. Studies by Dynan et al. [55,56,66] indicated that NonO and SFPQ are also two essential factors in NHEJ. The NonO:SFPQ heterodimer, along with the Ku complex, binds linear DNA fragments independently to form a functional preligation complex [56].

Matrin 3, a nuclear matrix protein initially proposed to stabilize selective mRNA species, is also involved in this early stage of the DSB response [67]. Matrin 3:NonO:SFPQ:Ku can also bind substrate DNA in vitro [56]. However, NonO:SFPQ only cooperates with Ku in *cis* (i.e., on the same DNA strand). Generally, NonO:SFPQ:Ku binds cooperatively to DNA substrates in vitro to form functional preligation complexes [56].

Li et al. [66] demonstrated that the attenuation of NonO expression increased the frequency of radio-induced chromosomal aberrations. They further observed that NonO knockdown produced a DSB repair-deficient, radiosensitive phenotype. These data suggest that a loss of NonO may deter the cellular response to irradiation and DSB repair [66].

NonO:SFPQ and Ku also bind DNA substrates independently but in different manners. Udayakumar et al. [68] observed that Ku promoted the capture of radiolabeled DNA in a concentration-dependent manner. While NonO:SFPQ also promoted the capture of radiolabeled DNA, an ~four-fold higher concentration was required for equivalent retention [68]. Furthermore, this group showed that, unlike Ku proteins, NonO:SFPQ can bind DNA substrates without free ends. This suggested that the NonO:SFPQ complex can capture DNA sequences and, in cooperation with Ku, stabilize a synaptic preligation complex. Also, NonO:SFPQ stimulates kinase activity and increases DNA-PKcs autophosphorylation [68].

PSPC1, which is highly similar to NonO and is a DBHS member, has yet to be reported as a key component in DSB repair. However, the knockdown of NonO can be rescued by overexpressing PSPC1 [69]. This suggests that PSPC1 can replace NonO in teaming up with SFPQ to participate in NHEJ.

## 7. NonO and SFPQ in Homologous Recombination (HR)

Even though NonO and SFPQ have been purified together and participate in DSB repair, NonO alone has also been found to stimulate NHEJ and suppress HR [70]. SFPQ, on the other hand, interacts directly with RAD51 to participate in HR [59]. RAD51 family members are key proteins involved in HR responses as well as in repair and genome stability [71,72]. RAD51 is an essential recombinase in both meiotic and mitotic homologous recombination [73]. NonO and SFPQ were copurified on a large-scale screen as RAD51 binding partners [74]. Subsequently, SFPQ was shown to directly interact with the RAD51D isoform and to mediate epistatic effects on cell viability. SFPQ also promotes sister chromatid cohesion and maintains chromosome integrity [59].

SFPQ was also shown to interact with RAD51 in a 1:1 ratio to modulate its HR activity [73]. Furthermore, SFPQ promotes strand exchange between ssDNA and DNA/RNA hybrids in a transcription-associated manner [73]. Finally, SFPQ stimulates RAD51-mediated homologous pairing and strand exchange under low-RAD51 conditions while inhibiting RAD51-mediated recombination when RAD51 concentrations are optimal [73]. 

The evidence is clear that SFPQ is necessary for HR repair during DSB damage as well as homology-directed repair and sister chromatid cohesion [59].

## 8. NonO and/or SFPQ Are Associated with DSB Repair Factors

Numerous experiments indicate that NonO and SFPQ are associated with DSB repair factors. For example, NonO and SFPQ are directly associated with topoisomerase I (TOPI), whose major function is to relax supercoiled DNA and alleviate DNA helical constraints [75]. The cleavage and preligation half-reactions of TOPI are unaffected by NonO:SFPQ, whereas the propensity of the enzyme to “jump” between separate DNA helices is stimulated [75]. This interaction was buttressed by the observation that a NonO:SFPQ:TOPI complex is pulled down by an antibody recognizing TOPII beta [76].

TOPI is bound to numerous cofactors, and one of them, TopBP1, has been reported to interact with NonO and SFPQ to repair laser-induced DNA damage sites [77]. NonO, possibly via its RRM domains, interacts with BRCT domains 6–8 of TopBP1, as demonstrated in a yeast two-hybrid system [77]. Considering that TopBP1 expression comes earlier (5 s, following DNA damage induction) than NonO and SFPQ (20 s), TopBP1 might recruit NonO and SFPQ to DSB sites. The three proteins reach their maximal concentrations at around 60 s. At some point, NonO and SFPQ disengage, while TopBP1 remains bound [77].

The Ku–DNA end-binding complex, which plays various roles with NonO:SFPQ as discussed previously, is involved in DSBs at the earliest stage [67]. Ku proteins bind to XRCC-DNA ligase IV and other NonO:SFPQ-interacting proteins to form “scaffolds” that stabilize DNA pairing (reviewed in [78] with particularly insightful Figures 1–3). NonO:SFPQ stimulates the autophosphorylation of DNA-PKs, and while the mechanism remains speculative, Udayakumar and associates (Figure 3 of [68]) provide an informative model that captures the essential points. 

As with their roles in HR, RAD51 proteins perform key functions in homologous recombination in DNA repair and chromosomal integrity (reviewed in [78]). SFPQ and NonO were among the four candidates pulled down in an attempt to identify the interaction profiles of RAD51D, whereas SFPQ was pulled down by RAD51C (illustrated in Figure 3 of [74]). SFPQ was subsequently shown to interact with RAD51D and participate in the homology-directed repair of DSBs [59]. The depletion of both SFPQ and RAD51 led to a lethal phenotype, whereas the reduced expression of SFPQ and RAD51D interrupted the cell cycle progression, leading to G2/M arrest and/or chromosomal aneuploidy. Thus, SFPQ directly participates in homologous repair in DSB DNA damage [59].

A series of parallel studies determined that SFPQ regulates RAD51-mediated homologous pairing between single-strand (ss) DNA and supercoil double-stranded (ds) DNA [73]. At low RAD51 concentrations, SFPQ promotes homologous pairing but inhibits such a pairing at concentrations in which RAD51 can function alone [73]. In the same paper, it was observed that SFPQ modulated RAD51-mediated strand exchange in a concentration-dependent manner, i.e., promoting strand exchange at low RAD51 concentrations while inhibiting the reaction when the RAD51 concentrations were optimal. SFPQ binding can compete with the ssDNA binding of RAD51 when its concentration is high, culminating in the disassembly of the RAD51–ssDNA interaction (schematic illustrated in Figure 3 of [73]). 

RAD9, the yeast ortholog of 53BP1, encodes an adaptor protein required for *S. cerevisiae* cell cycle checkpoint arrest in G1/S, intra-S and G2/M [79]. RAD9 also plays a role in the postreplication repair (PRR) pathway via the transmission of a checkpoint signal via the phosphorylation of the RAD9–HUS1–RAD1 (9-1-1) clamp complex (please see model in Figure 3 of [80]). While there is no evidence indicating that NonO or SFPQ interacts with 53BP1, the depletion of SFPQ delays DSB repair [67]. However, 53BP1 foci disappear when the expression levels of SFPQ are reduced [67].

Poly(ADP-ribose) polymerase-1 (PARP-1), one of several members of the PARP family, is a strong sensor of DNA damage (reviewed in [81]). PARP-1 rapidly produces Poly(ADP-Ribose) (PAR) [70], which appears to be involved in DNA damage repair (reviewed in [82]). NonO has been reported to be a PAR-binding protein [70], which colocalizes with PARP-1 and PAR at laser-IR-induced DNA damage sites immediately after the introduction of DNA lesions [70]. The recruitment of NonO to the DNA damage site is PARP-1- and PAR-dependent, and it is mediated via RRM1 of NonO [70]. Moreover, either the knockdown of NonO expression or PARP inhibition decreases NHEJ, while the attenuation of NonO not only decreases NHEJ but also facilitates repair by homologous recombination (HR) [70].

It has been speculated that NonO:SFPQ might be a good substitute for XRCC-like factor (XLF) or nonhomologous end-joining factor 1 (NHEJ1) to promote the sequence-independent pairing of DNA substrates in vitro. But it is clear that the ability of NonO to bind RNA contributes to DSB repair (please view their model in Figure 7 of [83]).

## 9. Posttranscriptional Modification of NonO Facilitates DNA Damage Repair 

Protein *O*-GlcNAcylation, catalyzed by *O*-GlcNAc transferase (OGT), has been linked with DNA damage (reviewed in [84]). The removal of OGlcNAcylation is catalyzed by *O*-GlcNAcase (OGA). OGA relocates to sites of DNA damage, where its C-terminal pseudo-Histone Acetyltransferase (HAT) domain plays a key role in its recruitment as well as its substrate recognition [85]. NonO and the Ku70/80 complex are *O*-GlcNAcylated by OGT [85]. A delay of OGlcNAcylation at DNA lesions delays NonO degradation and impairs NHEJ [85]. 

RING finger protein 8 (RNF8) is a major E3 ubiquitin ligase that, via its FHA domain, rapidly accumulates at sites of DNA damage via its FHA domain to facilitate the phosphorylation of MDC1. The phosphorylation of MDC1 occurs in response to DNA damage and is mediated by phosphoinositol-3-kinase-related kinases, mainly by Ataxia-Telangiectasia-Mutated (ATM) and Rad3-related (ATR) kinases [86,87]. NonO, but not SFPQ or PSPC1, is a substrate of RNF8 [87]. NonO is necessary for the loading of TopBP1 and ATR-interacting proteins (ATRIP) to chromatin following UV irradiation [87]. RNF8 also mediates the ubiquitination and degradation of NonO—an event necessary to terminate ATR-CHK1 checkpoint signaling by UV-induced DNA damage repair (UV-DDR) and required for S phase progression; please view Figure 7 of [87] for a proposed model of checkpoint signaling via NonO degradation in response to its RNF-mediated degradation. The mutation of three key NonO lysine residues prolongs the S phase after UV exposure [87]. 

These data suggest that the OGlcNAcylation and ubiquitination of NonO act in the absence of SFPQ to stabilize and retain UV-induced chromatin–protein complexes. Conversely, the degradation of NonO is necessary for cell cycle progression.

## 10. NonO and SFPQ in Telomere Stability

Neither NonO nor SFPQ are directly involved in the activity of telomerase [27]. However, both are pulled down in a complex with telomeric repeat-containing RNA (TERRA) [88], a class of long noncoding RNAs transcribed at telomeres that actively participate in regulating telomere maintenance and chromosome end protection. While the depletion of NonO or SFPQ in malignant cells does not impact the total TERRA levels, it does increase the TERRA foci per nucleus [88]. Further, a depletion/loss of NonO or SFPQ increases the RNA:DNA hybrid formation between TERRA and the C-rich telomeric strand [88]. 

The phosphorylation of both ATR and serine 33 of the 32 kDa subunit of replication protein A (RPA32pSer33) is a marker of replication stress [89]. A depletion of NonO and SFPQ results in the recruitment of RPA, and this effect can be abolished by the expression of RNaseH1 [88]. These observations indicate that NonO and SFPQ interact with TERRA to prevent the formation of RNA:DNA hybrids and R-loop replication defects at telomeric repeats [88].

Petite et al. (Figure 7 of [88]) have proposed an elegant model for the role of NonO and SFPQ in controlling telomere stability based on the composition of DNA, RNA and proteins [88,90]. Perhaps NonO and/or SFPQ are involved in either DNA/RNA or protein interactions that recruit other members to stabilize the telomere complex. This hypothesis is buttressed by the observation that NonO and SFPQ have been recently discovered to be regulators of telomere length homeostasis by suppressing telomere fragility and HR triggered by the TERRA-induced RNA:DNA hybrid. A loss of both NonO and SFPQ results in an increase in homologous recombination and altered telomere length homeostasis.

## 11. Emerging Role of NonO in Human Birth Defects

NonO is encoded on the X-chromosome, where hemizygous, cognitive dysfunctional variants were initially confirmed in NonO KO mice in 2015 by Mircsof et al. [91]. Since that time, human male NonO variants, both materially inherited or spontaneously arising, have been observed in over 20 families (recently reviewed in [92]) and are collectively referred to as “NonO-associated syndromic disorders (NASDs)”.

NonO pathogenic variants suffer from developmental delay, corpus callosum anomalies, muscular hypotonia, macrocephaly and a wide array of facial dysmorphisms.

Several of the recently confirmed afflicted children also suffer from epilepsy as well as hematologic problems, including a reduction in thrombocytes [92]. One child was found to have congenital aplastic anemia [53]. We find that this symptom is particularly instructive given the strong connection of NonO and SFPQ with cancer (discussed below).

A wide range of inherited and spontaneous genetic mutations underlie the broad array of defective genotypes. These include the deletion of the first six exons (mostly the 5′ noncoding region) and multiple deletions within coding regions (including, in one case, the complete skipping of Exon 7), which results in an in-frame deletion at the protein level and the corruption of the NOPS domain (Figure 3).

In summary, all children suffering from NASD demonstrate global developmental delay. Variants arise either de novo or from a carrier mother. Although carrier mothers are rarely or mildly afflicted, one mother presented with modest learning disabilities [93].

## 12. Implications for Human Health

This review has addressed select roles of two multifunctional proteins—the Non-POU-Domain-Containing Octamer-Binding Protein (NonO) and the Proline-and Glutamine-Rich Splicing Factor (SFPQ). We elected to focus on topics that have been less thoroughly reviewed, including NonO/SFPQ’s function in nonhomologous end-joining (NHEJ), homologous recombination (HR) and the repair of DNA double-strand breaks (DSBs). We attempted to summarize in these contexts how the absence or malfunction of NonO, SFPQ or both results in genome instability, apoptosis and cellular senescence. Finally, we assessed the roles of several NonO- and SFPQ-interacting proteins in these contexts, including Ku, PARP, TopBP1 and TERRA. Finally, we reviewed emerging and quite debilitating NonO-associated birth defects that appear to occur independently of SFPQ.

We find it interesting that NonO was initially isolated from a B cell leukemia [1]. Indeed, it was recently shown that the germline deletion of NonO in mice impaired B cell, but not T cell, development at the early pro- to pre-B cell level, resulting in the apoptosis of mature B cells [91]. While yet to be implicated clinically in B cell neoplasias, NonO is a key component in NHEJ and DSB repair, which are mechanisms that are critical for B- and T-cell receptor generation. During V(D)J recombination, DSBs induced by recombination-activating gene proteins (RAG1 and RAG2) are virtually all repaired by the NHEJ pathway for the benefit of antigen receptor gene diversity ([96]; reviewed in [97]).

While neither NonO nor SFPQ has been characterized further in lymphoma/leukemia, the aberrant functions of both have been observed in the etiology of colorectal, hepatocellular, renal, myeloid and prostate cancer ([98]; reviewed in [99]). Particularly, penetrant connects these two factors in neuroblastoma. Suitable treatments for low-risk patients exist, but high-risk neuroblastoma patients have exceedingly poor survival rates and lack therapeutic options. In particular, a high overexpression of NonO is associated with poor survival [100]. 

Recently, Zhang and colleagues [101] proposed a model for the regulatory role of NonO in neuroblastoma. They contend that NonO binding to the pre-mRNA of enhancer-regulated genes promotes the formation of RNA-processing paraspeckles to allow efficient splicing. Such a model supports the growing body of evidence of NonO and possibly SFPQ upregulation in different cancer cell types and clinical samples [34,48], suggesting that NonO might represent a potential therapeutic target.

## 13. NonO May Be Part of a Transcriptional “Super-Mediator” Complex

Beyond the observations made in neuroblastoma, NonO, through its association with SFPQ and other factors, may warrant consideration as a transcriptional supermediator. For example, although NonO is not a crucial component in spliceosome assembly, it interacts with critical spliceosomal proteins [47,102]. NonO also participates in a coregulatory network through promoter binding with ERK in stem cells [103] or by binding to metabolic gene pre-mRNA in hepatocytes [104]. Additional supermediator genes are regulated by NonO via diverse mechanisms, including GATA2, MYC and HAND2 [105,106,107]. Indeed, Zhang et al. [101] suggested that NonO acts as a “molecular scaffold” for HAND2, GATA2 and other regulatory contexts. These authors further showed that NonO interacts with HAND2 and GATA2 superenhancers from nonparaspeckle nuclear foci [101].

We find it interesting that the supermediator activity of NonO may function in neuroblastoma via its interaction and coregulation of BRD4/Bromodomain, which retards neuroblastoma growth via apoptosis [108]. NonO and BRD4 form nonparaspeckle nuclear foci [108,109]. 

As a clinical manifestation of the above observations, NonO inhibitors might be good prospects for inhibiting growth via the induction of apoptosis in neuroblastoma. A potential avenue of exploration should include small-molecule inhibitors of NonO function. The observations summarized in this report may provide a new approach to the development of pharmaceutical drugs to manipulate the aberrant RNA-binding capacity underlying cancer and other diseases.

## Figures and Tables

**Figure 1 jdb-12-00003-f001:**
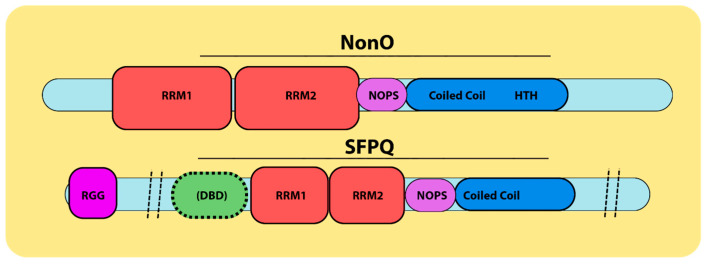
DBHS family nomenclature. NonO has also been termed the 54 kDa Nuclear RNA-Binding Protein (p54nrb) and the 55 kDa Nuclear Matrix Protein (nmt55). SFPQ was originally named the PTB-associated splicing factor (PSF). NonO and SFPQ are two members of the DBHS family. The DBHS conserved regions contain two RNA recognition motifs (RRMs, in red), a NONA/ParaSpeckle (NOPS) domain (in purple) and a coiled coil region (in blue). SFPQ also contains an N-terminal Arginine Glycine (RGG) region (in pink) and an uncharacterized DNA-binding domain (DBD, in green) at the N-terminus of its DBHS region. NonO has a highly charged helix–turn–helix (HTH in blue) C-terminal to its DBHS region, which has been suggested to have DNA-binding activity [19,20,21,22,24].

**Figure 2 jdb-12-00003-f002:**
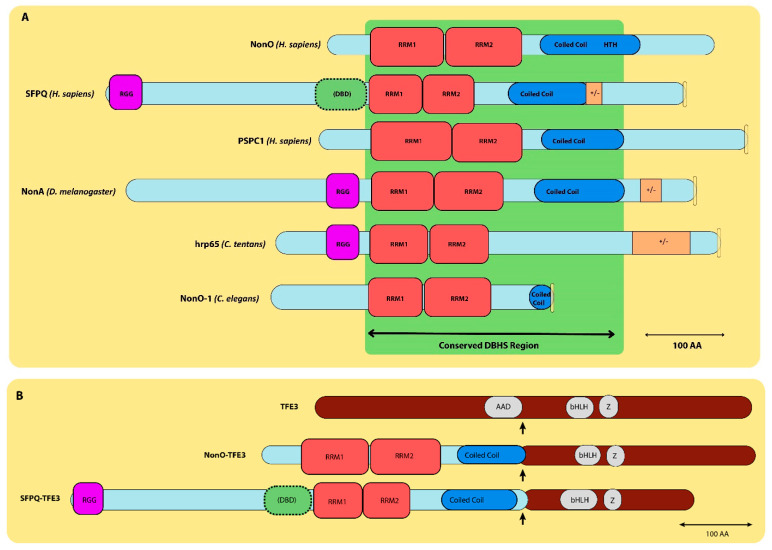
Alignment of DBHS family proteins. (**A**) Structural elements of DBHS family proteins: NonO, SFPQ, PSPC1, NonA, hrp65 and NonO-1a. Their C-termini contain the conserved DBHS region (shown in green) consisting of (from N-terminus to C-terminus) RNA-binding domains (RRM1, 2), NONA/ParaSpeckle domains (NOPSs, not pictured) and coiled coil domains. Also pictured are positively/negatively charged residue regions (+/−, brown), HTH domains (blue) and NLS regions (yellow). (**B**). NonO, SFPQ and TFE3 chimeric proteins. Structural representations of the wild-type NonO, SFPQ and TFE3 (transcription factor binding to IGHM enhancer 3) chimeric proteins. The TFE3 protein contains acidic activation (AAD) and DNA-binding domains (bHLH; Z) and has an overall length of 575 amino acids. Scales in (**A**,**B**) are indicated with scale bars representing 100 amino acids (aa) at the lower right.

**Figure 3 jdb-12-00003-f003:**
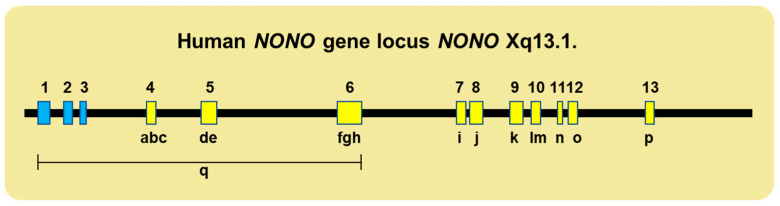
Various mutations across the human *NONO* locus are associated with birth defects. The top line shows, to scale, the coding (yellow) and noncoding (blue) exons of *NONO* located at Xq13.1. The numerals a–p denote the positions of pathologic point and/or deletion mutations within coding regions; q denotes the deletion of exons 1–3, which includes part of the 5′ noncoding region. Details of the various mutations and their clinical ramifications are provided in references [52,53,92,93,94,95].

## Data Availability

Not applicable.

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
