# Peer review of "Established and Evolving Roles of the Multifunctional Non-POU Domain-Containing Octamer-Binding Protein (NonO) and Splicing Factor Proline- and Glutamine-Rich (SFPQ)"

_jdb, 2024, doi:10.3390/jdb12010003_

Round 1
Reviewer 1 Report
Comments and Suggestions for Authors
Report on the review article by Yu et al. submitted to the Journal of Developmental Biology. In this short review the authors decided to focus on the specific function of SFPQ and NONO, two proteins of the DBHS family, and their involvement in DNA repair human diseases. The authors assemble a list of facts, reporting a number of articles. One can regret that they authors do not connect these facts together in order to provide an integrated view. I would recommend to add a couple of integrated figures in which the authors would provide a general scheme of SFPQ and NONO implication.
Minor comments:
Line 22: as a ?
Line 28: 3’-end pre-mRNA processing
Line 34: The DBHS family ..;
Line 35-36: This is too much simplification. The DBHS proteins are indeed composed of the cited domains but they are not all involved in DNA or RNA binding. This sentence should be split.
Line 40: X-ray
Line 40: There are more references to cite here.
Line 48: Please provide references for that.
Comments on the Quality of English LanguageThe overall seems correct although some sentences look a bit "poor" but I would let a motjher tongue person judge it.
Reviewer 2 Report
Comments and Suggestions for Authors
The authors of Yu et al review functions of DBHS family members NonO and SFPQ in the context of what is known for DNA repair and regulation by cell cycle and what is emerging with human mutations in these genes driving cancer and human disease functions. The review achieves these goals with a greater focus on DNA repair than cell cycle regulation. The human health section was insightful. The comments below would help with content, flow and readability of the review.
Major comments
-
Lines 34-37: it should be clarified in the main text which are found in humans/mice, which may be orthologs to which genes in lower species. Otherwise it seems like more than three genes are operating in the same species. The recommendation would then be to either highlight one of the lower species with their ortholog or remove them entirely since there is no further discussion of those genes.
-
Figure 1 seems redundant with Figure 2 and not clear if the NOPS domain is present on related proteins, given this is the conserved DBHS region across all genes. Perhaps it may be more useful to elaborate on amino acid conservation percentage across the conserved region.
-
Dimerization appears to be important but no description is provided about which domain(s) are involved. Could the authors expand on which domains are important for dimerization?
-
Figure 2 legend needs to describe the domains of TFE3. Fig2B seems to redundantly show NonO and SPFQ and should be removed.
-
Figure 3 is small and difficult to read the point mutations. Please enlarge; exon 2 needs to be shifted left and exon 10 is mislabeled. The genome build is not indicated and not clear what parts (dots/deletion) are purple; what then do the blue dots represent? In this regard, it is better to leave dots uniform and recommend further differentiation from Figure 2A of Ref 50 to minimize overlap and potential copyright issues.
-
It is not clear what the RGG domain is short for or putative function for SFPQ.
Minor comments:
-
As shown in the keywords, it would be best to introduce the alias names for NonO (NMT55) and SFPQ in the main text. Perhaps also PSPC1 (PSP1).
-
Page 1 line 34: NonO-1a? Figure 2 is described as such so either need to change for consistency.
-
RRM should be clearly defined in the main text in addition to the figure legend.
-
Figure color consistency is also problematic: RGG domain is not the same color.
-
Line 359 contains red text.
Round 2
Reviewer 1 Report
Comments and Suggestions for Authors
I still think that the purpose of a review is to summarize a number published manuscripts, and to this end, and especially in this review focused on DNA repair, it is almost mandatory to refer to an original (even original) Figure(s) that would recapitulate the role of DBHS proteins in DNA repair processes. This is still missing in that review.
The authors have made sporadic changes over the introduction section and last section only. In my opinion, this is not sufficient to substantially correct de various weaknesses of the manuscript. I would recommend to not accept the proposal.
Sentence line 37-40:
Still not very clear. The structures of DBHS proteins have been published and demonstrate that they are similar in their core. The end of the sentence is very not clear. I guess that the authors want to indicate that the dimerization is supported by the RRMs and the coiled coil regions, which as the most conserved domains of the proteins.
First paragraph of “DNA repair” section deals with “cell cycle” section. I guess that the authors wanted to link both processes and their connection via NONO:SFPQ but this is a bit confusing. Next, a paragraph is dedicated to DNA repair and the following section treats NHEJ, which is a DNA repair process. And again the following one is devoted to HR. And so on with DSB…
I guess there is a need to polish all that to get a clearer text building.
Figure 1 is now redundant to Figure 2A (top).
Figure 2A: NOPS regions are missing, they are not pictured as mentioned in the legend but it is not clear why not.
Minor comments:
Line 40: RRMs and not RRMS.
Line 101: repetition of”complex”
Line 132: international abbreviation for kilo is k and not K, so 200kDa and not 200KDa.
Line 145: binds
Comments on the Quality of English Language
none
